# Clinical Implications of Prominent Cortical Vessels on Susceptibility-Weighted Imaging in Acute Ischemic Stroke Patients Treated with Recanalization Therapy

**DOI:** 10.3390/brainsci12020184

**Published:** 2022-01-29

**Authors:** Misun Oh, Minwoo Lee

**Affiliations:** Department of Neurology, Hallym Neurological Institute, Hallym University Sacred Heart Hospital, Hallym University College of Medicine, Anyang 14068, Korea; minwoo.lee.md@gmail.com

**Keywords:** prominent cortical vessel, SWI, leptomeningeal collaterals, recanalization therapy, outcomes, stroke

## Abstract

Prominent cortical vessels on susceptibility-weighted imaging (PCV–SWI) correlate with poor leptomeningeal collaterals. However, little is known about PCV–SWI in recanalization therapy-treated patients with anterior circulation large vessel occlusions (LVO). We investigated PCV–SWI-based assessment of leptomeningeal collaterals and outcome predictions in 100 such patients in an observational study. We assessed PCV–SWI using the Alberta Stroke Program Early CT Score and evaluated leptomeningeal collaterals on multiphase CT angiography (mCTA). Predictive abilities were analyzed using multivariable logistic regression and area of receiver operating curves (AUCs). The extent of PCV–SWI correlated with leptomeningeal collaterals on mCTA (Spearman test, *r* = 0.77; *p* < 0.001); their presence was associated with worse functional outcomes and a lower successful recanalization rate (adjusted odds ratios = 0.24 and 0.23, 95% CIs = 0.08–0.65 and 0.08–0.65, respectively). The presence of PCV–SWI predicted outcomes better than good collaterals on mCTA did (C-statistic = 0.84 vs. 0.80; 3-month modified Rankin Scale (mRS) 0–2 = 0.75 vs. 0.67 for successful recanalization). Comparison of AUCs showed that they had similar abilities for predicting outcomes (*p* = 0.68 for 3-month mRS 0–2; *p* = 0.23 for successful recanalization). These results suggest that PCV–SWI is a useful feature for assessing leptomeningeal collaterals in acute ischemic stroke patients with anterior circulation LVO and predicting outcomes after recanalization therapy.

## 1. Introduction

Stroke is the second leading cause of death and the third leading cause of disability globally [1]. The pathophysiology of ischemic stroke is brain water homeostasis and inflammation regulated by the glial cells. Recent studies have shown that the glial water channel (aquaporin-4) and the glymphatic system could be involved in the pathophysiology of neurological disorders, including stroke, and implementing their therapeutic potential of stroke [2,3,4].

Endovascular thrombectomy for acute ischemic stroke (AIS) has become the first-line therapy for patients with an anterior circulation large vessel occlusion (LVO) [5,6,7,8]. However, a significant number of patients have poor functional outcomes despite successful recanalization after endovascular thrombectomy [5]. Therefore, performing advanced multimodal CT or MRI to assess collaterals, the ischemic core, and the penumbra is essential for identifying patients who are likely to benefit from recanalization therapy [7,8,9,10].

Good collateral circulation that maintains tissue perfusion after an LVO is one of the best predictors of good outcomes in recanalization therapy-treated AIS patients [11,12,13,14]. By extending the survival time of the penumbra, good collaterals can limit the expansion of the core infarction, the final infarction volume, and hemorrhagic transformation. Multiphase computerized tomography angiography (mCTA) is a reliable and non-invasive imaging tool for assessing leptomeningeal collaterals [15,16].

In clinical practice, susceptibility-weighted imaging (SWI) is a useful imaging tool for detecting an intracerebral hemorrhage, intra-arterial thrombus, microbleed, and hemorrhagic transformation of acute stroke [17]. Previous studies have shown that the extent of prominent cortical vessels observed on SWI (PCV–SWI), which identify the cortical pia mater, reflects the extent of hypoperfusion and is correlated with leptomeningeal collaterals in AIS [18,19,20]. Published studies have evaluated the value of PCV–SWI for assessing the status of collaterals and predicting outcomes in AIS; however, the clinical application of PCV–SWI for evaluating leptomeningeal collaterals prior to recanalization therapy remains inconclusive, since these studies generally have different study designs, patient characteristics, collateral-estimating imaging methods, and small sample sizes [20,21,22,23,24,25,26]. In particular, it is still unclear whether evaluating PCV–SWI can be considered as an alternative to mCTA for assessing leptomeningeal collaterals and predicting the outcomes of patients treated with recanalization therapy.

In the present study, we aimed to (1) test the value of PCV–SWI for assessing the leptomeningeal collateral status by comparing it with mCTA before recanalization therapy, (2) evaluate whether PCV–SWI can predict the outcomes of recanalization therapy-treated AIS patients, and (3) compare the accuracy of predicting outcomes using PCV–SWI relative to performing mCTA.

## 2. Materials and Methods

### 2.1. Patients

This was an observational, single-center, retrospective study based on a prospective stroke registry of 270 patients with AIS who were admitted to the Hallym University Sacred Heart Hospital and underwent recanalization therapy between January 2016 and December 2019. This study included consecutive patients who received recanalization therapy, had a pre-stroke modified Rankin Scale (mRS) score of 0–2, occlusion of the internal carotid artery (ICA) or middle cerebral artery (MCA) (M1/M2 segments), and underwent both SWI and mCTA. This study was approved by the Hallym University Sacred Heart Hospital Institutional Review Board. The need for informed consent was exempted.

### 2.2. Imaging Analyses

All patients underwent both a head/neck mCTA using a standardized protocol [15] and SWI using 3.0 T MRI. The extent of PCV–SWI was evaluated using the Alberta Stroke Program Early CT Score (ASPECTS), a 10-point quantitative topographic CT scoring system that subtracts one point from a maximum of 10 for each area with an abnormal signal intensity [18]. The PCV–SWI of the affected hemisphere was compared with the non-affected hemisphere and classified into two groups (Figure 1A): a PCV–SWI group (ASPECTS 0–7) and a no PCV–SWI group (ASPECTS 8–10) [18,22]. The leptomeningeal collateral circulation on mCTA (CC–mCTA) was evaluated using the imaging protocols of the Calgary Stroke Program of the University of Calgary, which classifies the leptomeningeal collateral status into two grades based on pial arterial filling: poor-to-intermediate (CC–mCTA score 0–3) or good (CC–mCTA score 4–5) (Figure 1B) [15].

Two observer-blinded stroke neurologists independently assessed the PCV–SWI and CC–mCTA results; any disagreements were resolved by discussion to reach a consensus. All imaging data were anonymized, and the reading of the scans was performed blinded to all demographic and outcome data.

### 2.3. Data Collection

Clinical data were collected on baseline demographics, medical history, and stroke characteristics, including age, sex, hypertension, diabetes mellitus, hyperlipidemia, atrial fibrillation, history of stroke, current smoking status, and type of recanalization therapy. The initial stroke severity was measured using the National Institute of Health Stroke Scale (NIHSS) score at admission, and the functional status at three months after stroke onset was assessed using the mRS score. Stroke subtypes were classified based on the Trial of Org 10172 in Acute Stroke Treatment (TOAST) criteria after the completion of diagnostic profiling [27]. Data were collected on the following variables: antiplatelet or anticoagulant administration, statin use before the index stroke, baseline glucose level, and baseline systolic blood pressure.

### 2.4. Outcome Measures

The primary clinical outcome was the three-month functional outcome measured using the mRS score during a regular clinical visit or through a structured telephone interview conducted by a trained research nurse. A good functional outcome was defined as a three-month mRS score of 0–2. The secondary radiologic outcomes were successful recanalization, defined as an expanded thrombolysis in cerebral infarction (eTICI) score of 2b or 3 [28], and any intracerebral hemorrhage (ICH) during hospitalization.

### 2.5. Statistical Analysis

Descriptive statistics are summarized as frequencies with percentages for categorical variables and as mean ± standard deviation (SD) or median (interquartile range, IQR) for continuous variables. Baseline characteristics were compared according to the presence of PCV–SWI. Outcomes were assessed using Student’s *t*-test or the Mann–Whitney U test for continuous variables, and the Pearson χ^2^, Fisher’s exact, or Cochran–Mantel–Haenszel shift test for categorical variables, as appropriate. The interrater reliability for evaluating PCV–SWI and CC–mCTA was assessed using unweighted k statistics. The relationship between the PCV–SWI and CC–mCTA was evaluated using the Spearman’s correlation coefficient.

Univariate and multivariate logistic regression analyses were performed using the backward stepwise method to determine the relationship between the PCV–SWI and the outcomes. To estimate the predictive accuracy of each imaging modality for outcomes, we developed separate logic regression models for each imaging modality as a predictor variable and calculated the accuracy of the outcome prediction. To compare the predictive accuracy and model fit of each imaging modality, we compared the C-statistics, Akaike information criterion (AIC), and Bayesian information criterion (BIC) derived from each multivariable logistic regression model. Multivariable models were created for age, sex, and variables that had univariate *p*-values < 0.05. Comparison of the area under the receiver operating curve (AUC) of the PCV–SWI (ASPECTS–SWI 0–7) and dichotomized CC–mCTA (good CC–mCTA scores of 4–5) was performed using MedCalc. To test the robustness of our findings, we performed additional sensitivity analyses using previously described models that were restricted to the patients who underwent endovascular therapy. Statistical analyses were conducted using R version 4.0.3 (R Foundation for Statistical Computing, Vienna, Austria) and MedCalc version 20.015. A *p*-value < 0.05 was considered statistically significant.

## 3. Results

### 3.1. Patient Characteristics

After excluding patients with a pre-stroke mRS > 2 (*n* = 5), with posterior circulation stroke (*n* = 53), without mCTA (*n* = 47), without MCA/distal ICA occlusion (*n* = 48), or without SWI (*n* = 16), we included 100 patients (mean age 70 ± 13 years; 54% men) in the study (Appendix A). The baseline demographic and clinical characteristics of the patients are summarized in Table 1. The median values of the ASPECTS–SWI and CC–mCTA scores were 4 (IQR 2, 6) and 3 (IQR 2, 4), respectively. There was a positive correlation between the total ASPECTS–SWI and CC–mCTA scores (Spearman’s rho = 0.77; *p* < 0.01) (Appendix A). The interrater reliability was excellent (ICC = 0.93; *p* < 0.01) for evaluating the collateral status and good (ICC = 0.89; *p* < 0.01) for the PCVs.

Among all patients, 67 (67%) patients had PCV–SWI and 33 (33%) did not. There were no significant differences between the two groups for baseline demographics, risk factors, baseline stroke severity, stroke subtype, or type of recanalization therapy. A good collateral status (CC–mCTA scores of 4–5) was less frequent in patients with PCV–SWIs than in those without them (25 vs. 82%; *p* < 0.01). The proportion of patients with good functional outcomes at three months was lower in patients with PCV–SWIs than in those without them (25 vs. 58%; *p* < 0.01), as was the proportion of patients who had successful recanalization (eTICI 2b or 3) (54 vs. 79%; *p* = 0.02). The proportion of patients with ICH was similar in both groups (28 vs. 24%; *p* = 0.66).

### 3.2. Clinical and Imaging Predictors Associated with Outcomes

Thirty-six (36%) patients had a good functional outcome at three months (mRS scores of 0–2). Patients who had a good functional outcome were younger (66 vs. 73 years; *p* = 0.03) and had fewer vascular risk factors, a lower median initial NIHSS score (13 vs. 16; *p* < 0.01), a lower median blood glucose level at admission (116 vs. 139 mg/dL; *p* < 0.01), and had fewer prescriptions of antithrombotic agents before the index stroke (28 vs. 48%; *p* = 0.04) (Appendix A). The distribution of occlusion sites also differed (*p* < 0.01). The imaging factors had a higher median ASPECTS–SWI (8 vs. 5; *p* < 0.01) and a higher median collateral score on mCTA (4 vs. 3; *p* < 0.01). PCV–SWI was significantly associated with a reduced probability of good functional outcomes after adjusting for age, sex, admission NIHSS score, admission glucose level, history of stroke, hypertension, prescription of antithrombotic agents, and location of the occlusion (adjusted odds ratio (OR) = 0.24, 95% confidence interval (CI) = 0.08–0.70; *p* = 0.01) (Table 2).

Successful recanalization (eTICI 2b or 3) was achieved in 62 (62%) patients. Patients with successful recanalization had a higher proportion of history of prior stroke (29 vs. 5%; *p* = 0.01) and a higher median ASPECTS–SWI (7 vs. 5; *p* < 0.01) (Appendix A). The presence of PCV–SWI was significantly associated with a reduced probability of successful recanalization after adjusting for age, sex, and history of prior stroke (adjusted OR = 0.23, 95% CI = 0.08–0.65; *p* < 0.01) (Appendix A).

Any ICH was observed in 27 patients (27%). Patients with ICH had a higher median glucose level at admission (147 vs. 124 mg/dL; *p* = 0.02) (Appendix A). After adjustment for age, sex, and admission glucose level, PCV–SWI was not associated with an increased risk of ICH (adjusted OR = 1.33, 95% CI = 0.50–3.55; *p* = 0.57) (Appendix A).

### 3.3. Comparison of Accuracy between PCV–SWI and CC–mCTA for Predicting Outcomes

Table 3 summarizes the C-statistic, AIC, and BIC derived from the multivariable logistic regression models, which used individual the imaging modalities as predictor variables. For predicting good three-month functional outcomes, the C-statistics from each logistic regression model of the ordinal ASPECTS–SWI, PCV–SWI (ASPECTS–SWI of 0–7), ordinal CC–mCTA scores, and dichotomized CC–mCTA (good CC–mCTA scores of 4–5) were 0.86, 0.84, 0.85, and 0.80, respectively (Table 3). The C-statistics for each of the imaging modalities were statistically significant. The AIC and BIC of the PCV–SWI were lower than those of the dichotomized CC–mCTA (AIC = 112 vs. 117, BIC = 135 vs. 140). Comparison of the AUCs revealed that the accuracy of PCV–SWI predictions was similar to that of the dichotomized CC–mCTA (*p* = 0.68) (Figure 2A).

For predicting successful recanalization, the C-statistics from each logistic regression model of the ordinal ASPECTS–SWI, PCV–SWI, ordinal CC–mCTA scores, and dichotomized CC–mCTA scores were 0.69, 0.75, 0.70, and 0.67, respectively (Table 3). The C-statistics of the ordinal ASPECTS–SWI, PCV–SWI, and ordinal CC–mCTA were statistically significant. The AIC and BIC of the PCV–SWI were lower than those of dichotomized CC–mCTA (AIC = 124 vs. 131, BIC = 137 vs. 144). Comparison of the AUCs showed that the prediction accuracy of the PCV–SWI was similar to that of the dichotomized CC–mCTA (*p* = 0.23) (Figure 2B).

For predicting any ICH, the C-statistics from each logistic regression model of the ordinal ASPECTS–SWI, PCV–SWI, ordinal CC–mCTA scores, and dichotomized CC–mCTA scores were 0.62, 0.61, 0.61, and 0.62, respectively (Table 3). None of the C-statistics of the imaging modalities were statistically significant. Comparison of the AUCs showed that the prediction accuracy of the PCV–SWI was similar to that of the dichotomized CC–mCTA (*p* = 0.60) (Figure 2C).

### 3.4. Subgroup Analysis

In the sensitivity analyses restricted to the 85 patients who underwent endovascular therapy, the multivariable logistic regression models showed that the PCV–SWI was significantly associated with a decreased probability of a good functional outcome (adjusted OR = 0.24, 95% CI = 0.06–0.87; *p* = 0.03) and successful recanalization (adjusted OR = 0.23, 95% CI = 0.08–0.65; *p* < 0.01) (Appendix A). There were no significant differences in the AUCs between the PCV–SWI and dichotomized CC–SWI (*p* = 0.77 for three-month mRS, *p* = 0.74 for successful recanalization) (Appendix A).

## 4. Discussion

The main findings of our study were as follows: (1) the extent of the PCV–SWI was correlated with the leptomeningeal collaterals assessed by mCTA in AIS patients with an anterior circulation LVO, (2) the presence of PCV–SWI was associated with an approximately 20% reduced probability of a good three-month functional outcome and achievement of successful recanalization in the recanalization therapy-treated AIS patients, and (3) the ability to predict outcomes using the PCV–SWI is similar to that of mCTA.

Our study showed that the presence of extensive PCV–SWI was inversely correlated with leptomeningeal collaterals on mCTA. This result supports the hypothesis that the extent of PCV–SWI can quantitatively reflect leptomeningeal collaterals [20]. In addition, patients with PCV–SWI had a lower rate of good collateral status, as assessed by mCTA, than those without them. Our results are in good agreement with those of previous studies that compared various imaging methods, including digital subtraction angiography (DSA) [18], dynamic perfusion-weighted imaging (PWI) [20], and multiphase MRI angiography collateral mapping [26] and showed a correlation between a pronounced presence of PCV–SWI and poor leptomeningeal collaterals. However, one study that used FLAIR and postcontrast time-of-flight (TOF) MRI angiography to estimate collaterals reported that extensive PCV–SWIs were associated with better collateral flow [19]. As discussed by Lee et al., this discrepancy could be caused by the difference in the imaging methods used to estimate the collaterals [26]. Collateral estimation using imaging methods without temporal resolution (FLAIR and post-contrast TOF MRI angiography) may lead to mislabeling of the leptomeningeal collaterals, as compared with collateral estimation using imaging methods with temporal resolution such as DSA, PWI, multiphase MRI angiography collateral mapping, and mCTA [26].

The postulated mechanism of PCV–SWI involves the ischemic tissue’s increased oxygen extraction fraction (OEF), which reflects the ratio of deoxyhemoglobin to oxyhemoglobin in the capillaries and veins. It is possible that an increased OEF shortens T2* relaxation and decreases the signal in the vessels of the affected hemisphere [17,29,30,31]. Therefore, it can be hypothesized that ischemic tissue with poor collaterals requires more oxygen, contains more deoxyhemoglobin in the vessels, and, thus, exhibits more PCV–SWI than that of tissue with good collaterals [32].

Previous studies have reported a varied relationship between PCV–SWI and functional outcomes in patients with AIS [22,23,24,26]. In a small study involving 40 AIS patients with an MCA occlusion, the presence of PCV–SWI was independently associated with a poor functional outcome at three months (OR = 55.77, 95% CI = 3.52–884.99), and the AUC for predicting a poor outcome was 0.78 (95% CI = 0.63–0.94) [23]. The most recent study involved 152 AIS patients, including 99 patients undergoing recanalization therapy for an MCA occlusion, and found a linear negative association between PCV–SWI and a good functional outcome at three months (*p* for trend = 0.008) [26]. In contrast, two other small studies on 22 AIS patients with an MCA infarction did not find a significant relationship between PVS-SWI (*p* = 0.34) and functional outcomes at three months [24] or one month [22] (*r* = −0.21 and *r* = −0.22, respectively). In our study, the presence of PCV–SWI was associated with significantly lower odds of a good functional outcome at three months (24%). In contrast to our study, previous studies included a mixed sample of patients in which some patients underwent recanalization therapy and some did not. For example, 63% of patients in the Wang study [23] and 65% of patients in the Lee study [26] underwent recanalization therapy; our study exclusively included patients who underwent recanalization therapy.

Although a previous study did not find a significant effect of PCV on achieving successful recanalization in a subgroup analysis of 25 patients who underwent recanalization therapy [23], our study showed that PCV–SWI was associated with a lower rate of successful recanalization. This result is consistent with reports that better collaterals are associated with a greater likelihood of successful recanalization [11,12,13].

In our study, the presence of PCV–SWI and collateral mCTA results each had good prognostic value for predicting functional outcomes at three months and successful recanalization in patients who underwent recanalization therapy. The predictive ability of the presence of PCV–SWI was similar to that of a good collateral mCTA result. The presence of PCV–SWI had a higher C-statistic and a lower AIC/BIC than a good collateral mCTA result, but the difference in the AUC was not statistically significant. Although mCTA is a practical imaging tool for making treatment decisions and predicting the outcome of recanalization therapy, it is not suitable for patients for whom contrast materials are not recommended. SWI does not use contrast materials and can be acquired easily using any MRI scanner. In addition, SWI provides additional information for assessing microbleed and detecting intra-arterial thrombi in patients with an anterior circulation LVO [17]. Our results suggest that PCV–SWI is a useful feature for collateral assessment and could replace mCTA, particularly in patients for whom contrast materials are not recommended.

In addition, our findings were replicated in subgroup analyses of patients who underwent endovascular recanalization therapy. None of the previous studies evaluated whether PCV–SWI could predict outcomes in patients who underwent endovascular therapy. The present study was the first attempt, and the presence of PCV–SWI could be deemed a valuable imaging feature for selecting patients who are more likely to benefit from endovascular therapy.

Our study has several strengths and limitations. This is one of the first studies to evaluate the value of PCV–SWI for assessing leptomeningeal collaterals and predicting outcomes that was conducted exclusively in AIS patients who presented with an anterior circulation LVO and underwent recanalization therapy. Thus, the present study specifically explored the role of using the PCV–SWI to guide recanalization therapy in AIS patients with an anterior circulation LVO. Second, our study is the first to use mCTA, a precise method that directly assesses collaterals, to verify the usefulness of the PCV–SWI for evaluating collaterals. The collateral score on mCTA, used to assess and quantify collaterals, is a widely accepted and validated scoring system that has been verified in a multicenter randomized clinical trial [15,16]. Our study is limited by its relatively small sample size and single-center design. Therefore, the sample of individuals who underwent recanalization therapy at our institution might have been biased; although, the acute stroke management was performed according to the institutional protocol, which is based on local and international guidelines. Second, not all the included patients underwent perfusion imaging under baseline conditions because our institutional protocol does not include perfusion imaging in pre-treatment imaging. Another limitation of our study is its retrospective design; however, the data were obtained from a prospectively collected registry. Further studies will be necessary to verify our results.

## 5. Conclusions

In conclusion, we found that evaluating PCV–SWI is feasible and useful for assessing leptomeningeal collaterals in AIS patients with an anterior circulation LVO and for predicting the outcome of recanalization therapy. Therefore, it should be considered as a principal feature in advanced MRI protocols for guiding recanalization therapy and identifying patients who are more likely to benefit from recanalization therapy.

## Figures and Tables

**Figure 1 brainsci-12-00184-f001:**
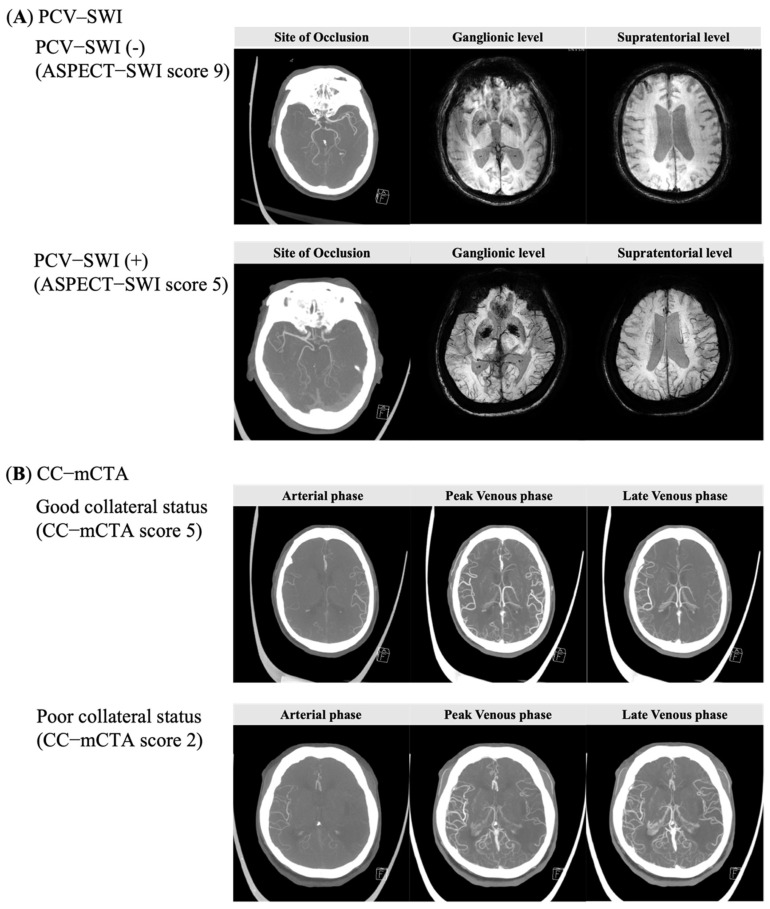
PCV–SWI (**A**), CC–mCTA (**B**).

**Figure 2 brainsci-12-00184-f002:**
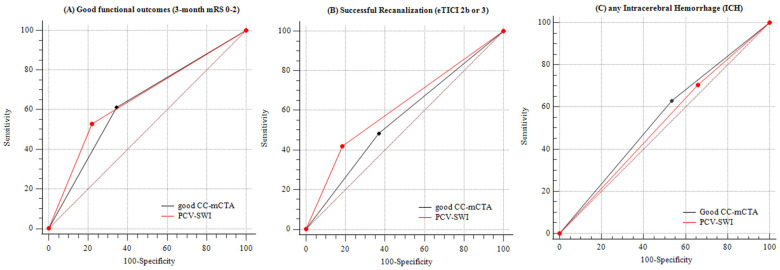
The comparison of AUCs between PCV–SWI and good collaterals–mCTA. (**A**) for predicting functional outcomes at 3 months, (**B**) for predicting achievement of successful recanalization, and (**C**) for predicting any intracerebral hemorrhage.

**Table 1 brainsci-12-00184-t001:** Patient characteristics, treatments, and outcomes according to the prominent cortical vessels on SWI.

	Patients, No. (%)			
	Overall (*n* = 100)	PCV–SWI (ASPECT–SWI, 0–7) (*n* = 67)	No PCV–SWI (ASPECT–SWI 8–10) (*n* = 33)	*p*-Value
Demographics and clinical characteristics				
Age, mean (SD), years	70 (13)	71.4 (13)	68 (13)	0.15 ^a^
Male sex	54 (54.0)	34 (50.7)	20 (60.6)	0.35 ^b^
Baseline NIHSS score, median (IQR)	16 (12, 18)	16 (13, 19)	15 (11, 18)	0.18 ^c^
Baseline SBP, median (IQR), mmHg	140 (130, 160)	140 (130, 160)	140 (129, 160)	0.43 ^c^
Baseline glucose concentration, median (IQR) mg/dL	130 (113, 160)	137 (113, 157)	124 (110, 169)	0.88 ^c^
Pre-stroke mRS				0.98 ^d^
0	90 (90.0)	61 (91.0)	29 (87.9)	
1	8 (8.0)	4 (6.0)	4 (12.1)	
2	2 (2.0)	2 (3.0)	0	
TOAST classification				0.35 ^d^
Large artery atherosclerosis	19 (19.)	9 (13.4)	10 (30.3)	
Cardioembolism	63 (63.0)	47 (70.1)	16 (48.5)	
Other determined or undetermined	18 (18.0)	11 (16.4)	7 (21.2)	
History of stroke	20 (20.0)	16 (23.9)	4 (12.1)	0.20 ^e^
Hypertension	70 (70.0)	50 (74.6)	20 (60.6)	0.15 ^b^
Diabetes mellitus	24 (24.0)	17 (25.4)	7 (21.2)	0.65 ^b^
Dyslipidemia	14 (14.0)	9 (13.4)	5 (15.2)	0.82 ^b^
Atrial fibrillation	57 (57.0)	42 (62.7)	15 (45.5)	0.10 ^b^
Current smoker	22 (22.0)	11 (16.4)	11 (33.3)	0.06 ^b^
Pre-stroke medication				
Antiplatelet or anticoagulants agents	41 (41.0)	29 (43.3)	12 (36.4)	0.51 ^b^
Statin	17 (17.0)	11 (16.4)	6 (17.2)	0.82 ^b^
Reperfusion therapy type				
IV thrombolysis	15 (15.0)	9 (13.4)	6 (18.2)	0.73 ^d^
Endovascular treatment	30 (30.0)	21 (31.3)	9 (27.3)	
Combined therapy	55 (55.0)	37 (55.2)	18 (54.5)	
Site of occlusion				0.45 ^d^
Middle cerebral artery				
M1	59 (59.0)	37 (55.2)	22 (66.7)	
M2	4 (4.0)	3 (4.5)	1 (3.0)	
Internal carotid artery	37 (37.0)	27 (40.3)	10 (30.3)	
ASPECTS–SWI, median (IQR)	4 (2, 6)	4 (3, 6)	9 (8, 9)	<0.01 ^c^
Collateral score–mCTA, median (IQR)	3 (2, 4)	3 (2, 4)	4 (4, 4)	<0.01 ^c^
Good collateral status (CC–mCTA, 4–5)	44 (44.0)	17 (25.4)	27 (81.8)	<0.01 ^b^
Outcomes				
Successful recanalization (eTICI 2b-3)	62 (62.0)	36 (53.7)	26 (78.8)	0.02 ^b^
Any intracerebral hemorrhage	27 (27.0)	19 (28.4)	8 (24.2)	0.66 ^b^
3-month mRS score				<0.01 ^b^
0–2	36 (36.0)	17 (25.4)	19 (57.6)	
3–6	64 (64.0)	50 (74.6)	14 (42.4)	

Abbreviations: PCV–SWI, prominent cortical vessel-susceptibility weighted image; SD, standard deviation; IQR, interquartile range; mRS, modified Rankin scale; NIHSS, National Institute of Health Stroke Scale; TOAST, Trial of ORG 10172 in Acute Stroke Treatment; LAA, large artery atherosclerosis; CE, cardioembolism; OE, other etiology; mCTA, multiphase CTA; ASPECTS, Alberta Stroke Program Early CT Score; eTICI, expanded Thrombolysis in Cerebral Infarction; mRS, modified Rankin Scale. ^a^
*p*-value by student’s *t* test, ^b^
*p*-value by chi-square test, ^c^
*p*-value by Mann–Whitney U test, ^d^
*p*-value by Cochran–Mantel–Haenszel shift test, ^e^
*p*-value by Fisher’s exact test.

**Table 2 brainsci-12-00184-t002:** Factors associated with good functional outcome at 3 months using logistic regression analyses.

	Unadjusted OR		Adjusted OR	
	OR (95% CI)	*p*-Value	OR (95% CI)	*p*-Value
Age	0.96 (0.93, 0.99)	0.02	0.98 (0.93, 1.03)	0.41
Sex (male)	1.88 (0.81, 4.36)	0.14	0.86 (0.28, 2.64)	0.79
Baseline NIHSS score	0.88 (0.81, 0.96)	0.01	0.89 (0.80, 0.99)	0.03
Initial glucose, mg/dL	0.99 (0.97, 0.99)	0.03	0.98 (0.97, 0.99)	<0.01
History of stroke	0.25 (0.07, 0.93)	0.04	0.25 (0.05, 1.36)	0.11
History of hypertension	0.23 (0.01, 0.57)	<0.01	0.32 (0.10, 1.01)	0.051
Prior antithrombotic agent	0.41 (0.17, 0.99)	0.046	0.79 (0.25, 2.54)	0.70
Site of occlusion		0.33		
M2 (reference)	-			
M1	2.90 (0.29, 29.5)	0.34		
Internal carotid artery	0.58 (0.05, 6.57)	0.66		
PCV–SWI	0.25 (0.10, 0.61)	<0.01	0.24 (0.08, 0.70)	0.01

Abbreviations: OR, odds ratio; CI, confidence interval; NIHSS, National Institute of Health Stroke Scale; PCV–SWI, prominent cortical vessel-susceptibility weighted image. Adjusted for age, sex, initial NIHSS score, glucose level at admission, history of hypertension, history of stroke, prior antiplatelet or anticoagulant agents, and prominent cortical vessels (PCV–SWI, 0–2).

**Table 3 brainsci-12-00184-t003:** Predictive ability of each imaging modality to discriminate outcomes using multivariable logistic regression analysis, receiver operating curve analysis, AIC, and BIC.

	Imaging Modality	Adjusted OR (95% CI)	*p*-Value	*C* Statistic	AIC	BIC
Good functional outcome (3-month mRS 0–2) ^a^	ASPECT-SWI (0–10), increase per 1 score	1.45 (1.16, 1.82)	<0.01	0.86	105	129
	PCV–SWI (≤7 versus >7)	0.24 (0.08, 0.70)	0.01	0.84	112	135
	CC–mCTA (0–5), increase per 1 score	2.04 (1.26, 3.30)	<0.01	0.85	108	132
	Good CC–mCTA (≥4 versus <4)	1.91 (0.71, 5.13)	0.19	0.80	117	140
Successful recanalization (eTICI 2b or 3) ^b^	ASPECT-SWI (0–10), increase per 1 score	1.23 (1.08, 1.51)	<0.01	0.69	132	145
	PCV–SWI (≤7 versus >7)	0.23 (0.08, 0.65)	<0.01	0.75	124	137
	CC–mCTA (0–5), increase per 1 score	1.49 (1.04, 2.13)	0.03	0.70	127	140
	Good CC–mCTA (≥4 versus <4)	1.65 (0.68, 3.98)	0.27	0.67	131	144
Any intracerebral hemorrhage ^c^	ASPECT-SWI (0–10), increase per 1 score	0.91 (0.77, 1.07)	0.27	0.62	124	137
	PCV–SWI (≤7 versus >7)	1.33 (0.50, 3.55)	0.57	0.61	125	138
	CC–mCTA (0–5), increase per 1 score	0.92 (0.64, 1.33)	0.67	0.61	125	138
	Good CC–mCTA (≥4 versus <4)	0.69 (0.27, 1.76)	0.44	0.62	124	137

Abbreviations: OR, odds ratio; CI, confidence interval; AIC, Akaike information criterion; BIC, Bayesian information criterion; mRS, modified Rankin scale; NIHSS, National Institute of Health Stroke Scale; PCV–SWI, prominent cortical vessel-susceptibility weighted image; CC–mCTA, collateral circulation score–multiphase CT angiography. ^a^ Adjusted for age, sex, baseline NIHSS score, baseline glucose level, history of hypertension, history of stroke, prior antiplatelet or anticoagulant use, and each imaging modality, ^b^ Adjusted for age, sex, history of hypertension, and each imaging modality, ^c^ Adjusted for age, sex, baseline glucose level, and each imaging modality.

## Data Availability

Anonymized data used in the current study will be shared through an appropriate request to the corresponding author.

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
