# Peer review of "Clinical Implications of Prominent Cortical Vessels on Susceptibility-Weighted Imaging in Acute Ischemic Stroke Patients Treated with Recanalization Therapy"

_brainsci, 2022, doi:10.3390/brainsci12020184_

Round 1

Reviewer 1 Report

Dear Editor,

The manuscript by Oh and Lee reports that Prominent cortical vessels on susceptibility-weighted imaging is a useful feature for assessing leptomeningeal collaterals in acute ischemic stroke patients with anterior circulation large vessel occlusions and predicting outcomes after recanalization therapy.

The design of the study and the technical quality of the work look somehow convincing and results can be of general interest. The authors have implemented correct statistical approach to analyse and present their data. However, there is a number of minor points that would need to be addressed in order to improve the quality of this paper before it can be accepted for publication:

Minor:

-The Introduction lacks a brief mention regarding the pathophysiology of stroke and essential role of glial cells in regulating brain water homeostasis. Glial cells, particularly astrocytes, appear to play critical and interactive roles especially at the BBB and BSCB. A recent breakthrough work by Kitchen et al Cell 2020, has showed that they are important targets for CNS disorders. These results have been confirmed by the work of Sylvain et al. who confirmed the role in stroke.

https://pubmed.ncbi.nlm.nih.gov/32413299/

https://pubmed.ncbi.nlm.nih.gov/33561476/

-Authors need to discuss the emerging role of glymphatic pathway which plays an important role in brain water homestasis. It is a waste clearance system that utilizes a unique system of perivascular channels, to promote efficient elimination of soluble proteins and metabolites from the central nervous system. Implementing this technique to study the glymphatic system will have huge therapeutic potential. References:

https://academic.oup.com/brain/advance-article/doi/10.1093/brain/awab311/6367770

https://pubmed.ncbi.nlm.nih.gov/30561329/

Author Response

Thank you for your thoughtful comments and suggestions in relation to our manuscript. Please see the attachment.

Reviewer 2 Report

Manuscript is well written. Methodology is robust.

Conclusions are appropriate. 

Author Response

We would like to thank your thoughtful comments and suggestions concerning our manuscript. Based on this beneficial information to improve our manuscript, we now submit a revision of the manuscript and responses to the reviewers.
